# Medicine Non-Adherence: A New Viewpoint on Adherence Arising from Research Focused on Sub-Saharan Africa

**DOI:** 10.3390/healthcare12080860

**Published:** 2024-04-19

**Authors:** Peter Michael Ward

**Affiliations:** Service Systems Research Group, WMG, University of Warwick, Warwick CV4 7AL, UK; p.ward@warwick.ac.uk

**Keywords:** medicine, adherence, sub-Saharan Africa

## Abstract

Adherence is vital for medicine to have an effect, yet adherence is considered to be low, with approximately half of the patients not fully adherent. However, research into adherence tends to focus on quantitative analysis of performance, which fails to perceive how people are adherent in their many different environments. As a contribution to gaining a deeper understanding, interviews were held with thirty individuals in the UK, Egypt, Kazakhstan, and six countries in sub-Saharan Africa to understand their perceptions on adherence to a range of drugs, and these were compared with an existing well-regarded list. New or undocumented reasons for non-adherence were discovered. Reasons for non-adherence were consistent across both developing and developed worlds. A new viewpoint on adherence is suggested, which considers adherence to be a single act and therefore as an individual opportunity to be adherent, permitting greater focus on the enablers and inhibitors of adherence at any given point in time.

## 1. Introduction

In his seminal 2003 report for the World Health Organisation (WHO), Sabaté [1] (p. xiii) said, “[Increasing adherence] may have a far greater impact on the health of the population than any improvement in specific medical treatments”. Adherence to instructions for medicine consumption is a fundamental requirement for health. Indeed, McColl-Kennedy et al. [2] refer to it as “Comply[ing] with basics”, yet non-adherence is a significant worldwide issue. For example, it has been estimated that 125,000 people die each year just in the USA as a result of non-adherence [3]; figures for other parts of the world are not known. In the developed world, half of the patients are not fully adherent to their prescription instructions [1,4,5], and it is thought that the proportion of non-adherence is higher in the developing world [1].

A significant amount of practical research has been performed on the issue of adherence [1,5]. Peterson et al. [6] found 95 studies on adherence. More recently, a narrative review [7] identified a total of 38 systematic literature reviews of adherence papers. A recent search of the MEDLINE database for the term “medicine adherence” revealed that almost 19,000 such papers have been published.

Sabaté’s World Health Organisation report is a milestone in the field. Building on his work, another empirical report, “Adult Meducation: Improving Medication Adherence in Older Adults”, produced jointly by the American Society on Aging and the American Society of Consultant Pharmacists Foundation [8], categorised 55 causes of non-adherence using the five “dimensions” of Sabaté’s report: health system/HCT, social/economic, therapy-related, and patient-related and condition-related factors; see Figure 1.

There are limitations to the practical research performed so far. Firstly, most research has had a primarily Western focus and may not be completely applicable in the developing world. Secondly, there has been a concentration on age-related issues in the USA and HIV/AIDS-related issues in sub-Saharan Africa. It is, therefore, possible that further important information on the causes of non-adherence, including details that may be specific to particular medicines or be geographically localised, still remains to be captured.

This study investigates people’s experiences of adherence in their lived lives, with the aim of exploring reasons for non-adherence and identifying new causes not documented so far. A series of semi-structured interviews was arranged with people who were willing to talk about their past experiences of taking medicines. They were located in various environments ranging from a comfortable urban environment in a developed country through to an impoverished rural environment in a developing country.

## 2. Materials and Methods

Interviewees were selected using purposive and snowball sampling [9]. Initial interviews were performed with six contacts in the UK to explore the situation in the developed world. Following that, twenty-four interviews were arranged with contacts in Kenya, Tanzania, Zambia, Zimbabwe, Uganda, Nigeria, and Kazakhstan. These were intended to explore the developing world, primarily sub-Saharan Africa. A total of thirty interviews were conducted over a period of just over five months from the end of December 2014 to early June 2015.

Semi-structured interviews were performed, either face-to-face or by telephone. Interviews generally lasted for 25–30 min. Table A1 and Table A2 in Appendix A summarise the interviewees. All interviews were performed by the author in English. All those asked were willing to be interviewed and gave their approval via reading a Participant Information Leaflet and agreeing to the terms of a Consent Form. The questions asked are listed in Table 1.

Each interview was recorded and transcribed. A combination of Nvivo and manual means was used to code the transcripts. The general approach of Systematic Combining [10,11] was used to revise the initial framework based on empirical findings. Codes were analysed and a taxonomy of non-adherence was created. Further analysis was performed to compare the reasons for non-adherence discovered in interviews with the list of 55 reasons from the “Adult Meducation” report [8].

## 3. Results

### 3.1. Coding Categories

Table A3 and Table A4 in Appendix B show phrases extracted from interview transcripts and how they were coded, looking separately at the developing and developed worlds.

Some examples of coding are as follows:Interviewee EG01 said, “…pharmacies in every street… just down the road from our flat”, and this was counted as “Distance, Positive, Close”, while interviewee UG01 said, “It’s 30 km to and from, to the pharmacy. USD 10 [GBP 6.57] transport” which was considered to be “Distance, Negative, Far”Interviewee NG01 said, “Sometimes I’ll take it according to the prescription but sometimes I stop when I feel better”, which was coded as “Stop, Negative, Better”, while interviewee KN03 said “They act like emergency for my family” which was coded “Stop, Negative, Keep”Interviewee KN08 said, “This tablets are in large sizes and so swallowing becomes a problem”, coded as “Size, Negative, Big”.

In this way, all relevant interview statements were captured and coded. Table 2 shows the coding derived from the interviews. As can be seen, not all categories have positive as well as negative attributes, but the focus of the interviews was on non-adherence and so this is to be expected.

As part of this work, surprises were found regarding the overall approach to adherence on the part of some interviewees. For example, some stopped taking medicine when they felt better even if it was an antibiotic; many struggled with tablets being too big to swallow or possessing a bitter taste; one commented on how the pharmaceutical industry was making profits from medicines; several were afraid of rumoured side effects. There was a wide spread of reasons for why adherence was not achieved.

### 3.2. Taxonomy

It proved possible to consolidate these reasons. Further analysis was performed to create a taxonomy of non-adherence categories, identifying five entities relating to non-adherence. Table 3 summarises this.

In line with normal usage, in this analysis, “agency” refers to the capacity of individuals to have the power to fulfill their potential, “affordance” is a property of an object that determines how it might be used, and “context” is the situation within which something exists or happens.

This taxonomy shows that motivation is just one cause of non-adherence, despite being the one that receives strong focus. There are more reasons for non-adherence relating to the medicine than there are to the patient, while the consumption context is critical to adherence. Summarising this, from Table 3, it can be seen that there are three factors at play in adherence: patient, medicine, and context.

### 3.3. Reasons for Non-Adherence

As well as identifying these three factors, the reasons given for non-adherence were assessed against the list of 55 in the “Adult Meducation” report [8]. Ten causes in the report were not mentioned in the research. These were of the type where the interviewee would have to expose themselves to what may be considered an unacceptable degree or which needed to be inferred by the interviewer in a face-to-face situation. Examples are “Mental retardation” or “Alcohol or substance abuse”.

Table 4 shows the 19 reasons for non-adherence discovered in interviews which were not mentioned in the report [8]. While some of these might be obvious and anecdotally known, they have not been documented in formal research to date.

Similar causes of non-adherence were seen in both the developed and developing worlds. For example, a lack of food and water for taking tablets was referenced in both, yet this was not mentioned in the list of 55 causes. This suggests that interviews are of significant importance both to understand non-adherence reasons in detail and also to expand the list of known reasons.

## 4. Discussion

The qualitative research results have provided a rich view of adherence as part of people’s lived lives in a range of environments from extreme poverty to relative comfort, across both developed and developing worlds. The results have extended our understanding of the phenomenon of non-adherence and provided insights into the range of causes beyond prior knowledge.

### 4.1. Broadening the Scope of Adherence Research

The categories derived from the interviews provide a valuable picture of the broad spectrum of factors which make up adherence in context. The taxonomy of entities leads to the conclusion that to understand adherence, we must consider the three aspects of patient, medicine, and context together. It has not previously been normal to bring all three of these into research at the same time.

For example, it is clear that motivation is an important part of adherence, yet it is just one factor among very many. The focus on increasing motivation in a lot of adherence interventions is potentially missing the wider perspective. Even simply considering patient agency and beliefs broadens the scope of intervention. Based on this research, considering agency as relevant to adherence would bring into view the topics such as the length of a course, the imposition of the regimen on the patient’s routine, and the causes of stopping. Would it be possible to shorten the course or to reduce the number of doses per day? This would be an intervention on the product side which reduces the need for patient agency, thereby facilitating adherence.

Taking context and medicine into account could make an even more significant impact. Consumption context is a potential major area of investigation. This research identified seven categories of causes of non-adherence under the heading of context (Table 3): people, utensils, reminder, water, food, storage, and norms. Norms is a large area, raising questions of culture that then includes the effects of stigma on medicine consumption. But the issue of utensils, for example, could simply be addressed by providing a suitable spoon with the medicine.

The medicine itself is perhaps the area that could generate the largest potential improvement in adherence. Product affordance was a factor in thirteen categories of non-adherence including taste, size, and smell (Table 3). These could be addressed relatively simply by manufacturers if they were to take the issues seriously. Others might be more challenging but taking them seriously as causes of non-adherence could pay dividends.

### 4.2. Non-Adherence Reasons

The “Adult Meducation” report [8] documented 55 causes of non-adherence. This research uncovered 19 more. Many causes were seen in both developing and developed worlds, indicating that although root causes of non-adherence might be different in some cases, their manifestations are the same, for example, a lack of water, a lack of food, keeping medicine for future use, or misunderstanding the instructions.

Some causes of non-adherence would not routinely be considered in the developed world, for example, a dislike of supporting the pharmaceutical industry’s profits, or concern that the medicine is foreign. However, it makes sense to consider shared causes because interventions might be globally valuable or make a particular contribution to poorer areas, such as keeping medicine for future use or for family needs. This implies that price and availability are relevant, but also, in consideration of “feeling better”, a lack of understanding that some medicines must be consumed until the prescription is complete. As well as patient education, this implies the importance of providing clear instructions in a language that the patient understands and that is consistent in both written and verbal forms.

It may be seen that some of the factors of non-adherence are interrelated and can be traded off against each other. For example, if the affordance of the medicine is perceived by the patient as being inadequate in itself to permit adherence to take place, they may be able to call on other resources from context and agency to overcome such inadequacy. If the medicine is bitter, then the patient may be able to use their agency to bring sugar into context to sweeten it. If it requires food to be eaten at the time of consumption and there is none available, then support may be obtained from an alternative source. These simple examples demonstrate the potentially complex interactions between adherence factors.

Some adherence factors are effectively “mirror images” of each other. For example, a patient’s context may not be contributing sufficient resources to permit adherence, but if the medicine’s affordance were to be enhanced then consumption might still be able to occur. Perhaps a patient’s context cannot provide food or water, but if these could be incorporated into the medicine in some way then the patient may still be able to be adherent. Similarly, the patient’s agency may be limited—perhaps not being able to open the bottle or swallow large pills—but enhancements to the medicine might address such limitations.

### 4.3. Unit of Analysis of Adherence

One important facet of this research is the focus on adherence as an individual act rather than an average of all adherence events for a single patient or even a cohort of patients. This approach has highlighted reasons for non-adherence rather than just measuring it.

A lot of research on intervention highlights the limited impact that interventions achieve. For example, when van Dulmen et al. [7] reviewed 38 systematic reviews, they discovered that only 45% of interventions resulted in improved adherence, and only 33% in improved outcomes. Many papers discuss the need for, or evaluation of, multiple forms of intervention to improve adherence rates. This is discussed in two reviews [6,12]. Kardas et al. [12] suggested in their review that “multifaceted interventions may be the most effective answer”, but at the same time, they found that many of the reviewed papers reported mixed or limited success (for example [13,14,15]). Without an understanding of adherence enablers and inhibitors in patients’ lived lives such as has been discerned in this research, it is not surprising that interventions have limited impact.

### 4.4. Intention and Reality

When adherence research incorporates a theoretical perspective, it tends to use expectancy-value models, usually the Theory of Planned Behaviour [16,17], for example [18,19]. The limitation of such theories is that they reach only as far as the intention to act. They hold an implicit assumption that intention leads directly to behaviour, overlooking the possibility that it is not always true. This research has demonstrated that motivation—the intention to act—is just one element of adherence and that there are many factors that can prevent it, including those relating to the medicine and operating within the consumption context. A new theory of medicine adherence is required which recognises this in order to make progress towards higher adherence levels.

## 5. Conclusions

### 5.1. The Triad of Adherence

It is normal in adherence research to consider dyads. There is the dyad of prescriber and patient, for example. But this research has brought out the importance of considering the whole picture of the triad of the patient and medicine in a consumption context. Looking at all three aspects allows the full picture of adherence to be seen. Understanding the three aspects and how they interact with each other as a system provides insights into reasons for non-adherence that cannot otherwise be discerned. This approach has uncovered new reasons for non-adherence.

### 5.2. Reasons for Non-Adherence

Nineteen new reasons for non-adherence were documented as a result of this qualitative research. At a time when much of the adherence research is quantitative, assessing adherence by percentage compliance with instructions, it is important to understand that people have multiple reasons for their non-adherence which cannot be captured quantitatively. If we are to help people to become more adherent, we need to understand their circumstances. Putting all non-adherence down to a lack of motivation misses the point that this is just one of many facets. A deeper understanding of people’s lived lives can identify interventions which might make a difference to compliance.

Reasons for non-adherence were remarkably consistent across the developing and developed worlds. Though caused differently, the outcomes were the same. For example, a lack of water at the time of consumption was identified in both sub-Saharan Africa and the UK as a cause of non-adherence.

### 5.3. Adherence as a Point-in-Time Opportunity

Considering all this, it can be seen that adherence is not a percentage figure but is achieved or otherwise each time consumption is due. It is either 100% or 0%. Understanding the point-in-time reasons for non-adherence will permit actions to be taken which increase the number of times when adherence is achieved, thus enhancing the effectiveness of interventions.

For example, sometimes water is not available and adherence cannot be achieved. Reformulating the medicine so that water is not a corequisite will address this cause of non-adherence. It may only be effective one time in ten but at that time it makes a 100% difference in adherence. Viewing adherence as a percentage of all consumption opportunities may overlook this point.

### 5.4. Learning for the Pharmaceutical Industry

The points mentioned above suggest that medicine formulations might be more intelligently designed, and that this might benefit people worldwide. A lack of water to consume a tablet in Kenya might be due to there being no water in the well, but a lack of water in the UK could be that the patient is a passenger in a car. Whatever the cause, non-adherence is the result. What steps can be taken to remove the requirement of water from the consumption context? Can the medicine be provided in another formulation, perhaps? Can water be provided with the medicine? The first question relates to the manufacturer, while the second could be answered at the pharmacy. They could be long-term and short-term answers or could depend on the medicine.

Considering some of the other reasons for non-adherence, we might apply the same line of thinking to the subject:Lack of food: Can food be provided with the medicine? Can the active ingredients be incorporated into some form of food?Bad taste: Can the medicine be sweetened in some way? Can the taste be masked?Large size: Can the tablet size be reduced? Can the formulation be changed?Bad smell: Can the formulation be changed? Can the smell be masked?Lack of dosing spoon: Can a spoon be provided in the medicine packaging or by the pharmacist? Can the formulation be changed?

Considering the other categories identified, it seems reasonable to explore what the pharmaceutical industry can do to address medicine affordances in all the identified areas of content, branding, effects, taste, formulation, size, smell, instructions, regimen, distance, access, cost, and diagnosis. It may contribute to some of the contextual categories of people, utensils, reminder, water, food, storage, and norms. In particular, medicines which more completely address contextual challenges could be more successful in raising adherence than those which at present might be perceived as “one size fits all” or even “lowest common denominator”. Some factors will prove to be out of the manufacturers’ scope and perhaps more related to healthcare providers and pharmacies, but others might be easily tackled once they become the subject of some analysis.

Patient centricity is a goal for many in the industry, and taking this approach could enhance that focus. Using the insights gained from in-depth qualitative research could deliver new ways of supporting patients to be adherent, moving towards the goals of increased adherence and higher quality of life.

### 5.5. Research Limitations

The research was performed remotely. A more ethnographic approach might have both confirmed the remaining 10 causes of non-adherence present in the “Adult Meducation” report [8] that were not found in the research, and potentially uncovered additional causes through observation and interviews with family members, medical staff, etc.

Interviews in some countries were limited to just one. Further information may have been obtained with a greater number of interviewees per country.

This research considered only one developing country, the UK. Although this was not a focus of the research, which primarily addressed reasons for non-adherence in sub-Saharan Africa as a representative area of the developing world, investigation in other developed countries might have provided a richer picture of non-adherence reasons.

### 5.6. Opportunities for Further Research

It would potentially be useful to perform further qualitative research face-to-face with interviewees in their contexts. This should reveal a greater depth of insight and add further understanding of non-adherence in sub-Saharan Africa.

The same approach could be taken to explore adherence to products other than medicine. For example, a fitness regime or a smoking cessation course also requires the participants to be adherent. Considering adherence as a point-in-time opportunity would allow researchers to study the triad of the patient and the “product” in context to understand non-adherence in more detail.

Theoretical work on the development of a theory of adherence could pay dividends in increasing adherence. It would start from the position of recognising the complex dynamics operating between the elements of the triad of adherence and go beyond the focus on motivation to consider the holistic picture. Viewing adherence as a (complex) process where patient agency and medicine affordances come together into a consumption context would permit a deeper understanding of the interactions of the non-adherence categories in enabling or preventing adherence [20].

## Figures and Tables

**Figure 1 healthcare-12-00860-f001:**
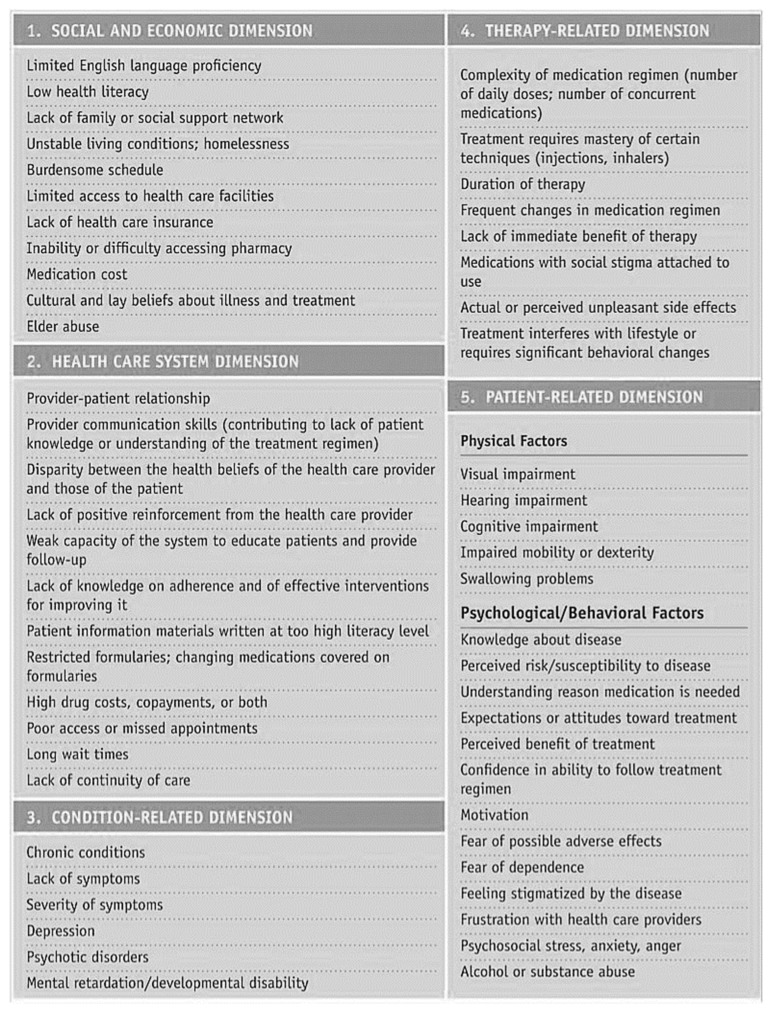
The 55 causes reported to affect adherence [8]. Table republished with permissions from the American Society on Aging and American Society of Consultant Pharmacists Foundation.

**Table 1 healthcare-12-00860-t001:** Semi-structured interview questions.

Number	Question
1	What medicine do you wish to share your experiences of?
2	Is this your first time with this medicine or is it a repeat prescription?
3	How far was it to a pharmacy?
4	How much did it cost you to buy the medicine?
5	Did you obtain the medicine?
6	If you obtained the medicine, how did you feel about it at the time?
7	Did you actually plan to consume it in line with the prescription?
8	Did you know how to take this medicine? How do you know?
9	Please describe your physical surroundings on various occasions when the prescription said you should consume. Who and what was there and not there?
10	What were you thinking and feeling?
11	How were your physical and mental health?
12	Did you actually consume at that time?
13	What helped you to consume or prevented you from consuming?
14	Is there anything about the medicine that makes it hard for you to take it? What would make it easier for you?
15	If you had the choice, how would you like to take this medicine?
16	Anything else you want to say about what makes it easy or difficult to take medicines for you personally?

**Table 2 healthcare-12-00860-t002:** Coding of interviews grouped by category.

Category	Positive Attributes	Negative Attributes
Distance	Close	Far
Access	Easy	Hard
Cost	Low	High Herbal, low
Diagnosis		Foreign language, verbal
Instructions	Clear, verbal Clear, written	Foreign language, verbal Foreign language, written Unclear, verbal Unclear, written Misunderstood
Utensils		Missing
People	Present	Absent
Content		Unknown
Norms		Others, stigma
Branding	Known	
Beliefs	Others Confidence	Others, too dependent Lack of faith Foreign origin Profit, pharma Profit, herbal Value Pointless
Motivation	Last resort Stay well Get well	Tired
Stop		Keep Replaced by other Discarded Better Busy Run out
Effects	Others Side, none	Others Side, general Side, specific Bad
Taste	Sweet	Bad Bitter
Formulation	Tablet Liquid Injection	Injection
Regimen	Acceptable	Unexpected Unacceptable Complex Forgot
Reminder	General Alarm	
Water	Present	Absent
Food	Present	Absent
Size	Small	Big
Smell		Bad
Course	Acceptable	Long
Routine	Present	Absent
Storage		Unsafe

**Table 3 healthcare-12-00860-t003:** Taxonomy of categories of non-adherence.

Taxonomic Entity	Categories
Patient motivation	Motivation
Patient agency	Course, routine, and stop
Patient beliefs	Beliefs
Consumption context	People, utensils, reminder, water, food, storage, and norms
Product affordance	Content, branding, effects, taste, formulation, size, smell, instructions, regimen, distance, access, cost, and diagnosis

**Table 4 healthcare-12-00860-t004:** Reasons for non-adherence beyond those documented in “Adult Meducation” [8].

Reason	Seen in Interview
Concern with medicine content	EG01
Verbal instructions in a foreign language	EG01
Written instructions in a foreign language	EG01
Pharmaceutical industry profits	EG01
Herbal medicine industry profits	EG01
Feeling better	KN03 UK05 TZ01
Lack of food	KN03 KN04 TZ01
Lack of water	KN08 UK01
Concern that medicine is of foreign origin	NG01
Lack of faith leading to need for medicine	TZ02
One medicine being replaced by another	KN03
Medicine kept for future occasions	KN03 NG01 TZ01 UK05
Medicine kept for family need	KN03 NG01 TZ01
Instructions misunderstood	UK01 KN05
Difference between written and verbal instructions	KZ01
Lack of routine	UK01
Lack of safe storage	TZ04
Forgetfulness	KZ01 TZ03
Run out of medicine	UK04

## Data Availability

Data are contained within the article.

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
