# Peer review of "Medicine Non-Adherence: A New Viewpoint on Adherence Arising from Research Focused on Sub-Saharan Africa"

_healthcare, 2024, doi:10.3390/healthcare12080860_

Round 1

Reviewer 1 Report

Comments and Suggestions for Authors

Authors need to look into these points for improvement of manuscript;

1) Abstract need to be refined with significant results and conclusions

2) Authors have concluded that reasons for non adherence were similar in both developed and developing world. This should be further explained with some examples and one example of water isn't sufficient. 

3) What is reason of selection of most subjects from Village side? Normally there are less facilities there so definitley results can be as expected or desired?

4) Only one subject is chosen from 5 countries which is not acceptable for any prediction of results. Reasons ?

5) Only one subject chosen of age below 20 years? 

6) Only one developed country UK is selected with only six volunteers so data cant be corelated technically

Comments on the Quality of English Language

English language be proofed again

Author Response

Thank you for your analysis of my paper. I have made the changes you request. I have annotated your analysis in the attached document. I trust that my responses to your analysis are satisfactory.

Reviewer 2 Report

Comments and Suggestions for Authors

The article “Medicine non-adherence: a new viewpoint on adherence arising from research in sub-Saharan Africa” is very interesting. I have following comments/suggestions,

1.      The abstract is clear.

2.      The introduction provides sufficient background in the topic and the aims are clear.

3.      Why is the author reporting the findings almost 9 years after the interviews were held?

4.      The IRB approval number is missing.

5.      The results are clear.

6.      The Discussion and Conclusion are in line with the results.

Author Response

Thank you for your analysis of my paper. I appreciate your encouragemetn. I have annotated your analysis in the attached document. I trust that my responses to your analysis are satisfactory.

Reviewer 3 Report

Comments and Suggestions for Authors

The title is research in sub-Saharan Africa, but the data includes other countries.  I would suggest to either retitle or to eliminate the data that is not from sub-Saharan Africa.

Watch using that and also.  Do not start a sentence with the actual number, you will need to spell out.  

Abstract needs significant improvement to be more reflective of the research work. Abstract contains a run on sentence.  Watch using However through paper and abstract.  Yet adherence is considered to be low--provide more context to this statement. 

Introduction line 26 stay with adherence rather than compliance

Line 29 sentence does not make sense ?much practical research? 

Line 30 a review of reviews (systematic review or narrative review)

Line 31-33 You can't conclude that the area is expanding just because there are 19000 papers.  Also, just because Medline has a hit of 19000 does not mean the focus of these papers was adherence.  It could be a clinical study that documented adherence as part of the study protocol.  Be very careful with statements like this. 

Line 40 remove However

Lines 55-57 should be moved to the discussion section

Materials and Methods

Data from 2014 -2015?  delay in publishing??

Line 61,  tables in appendix include zimbabwe and zambia (not in methods)

The materials and methods section needs to be expanded and more detailed to how interviews were conducted, how individuals were trained, did they all ask the same questions??  Need more details on the coding and analysis.

Line 75 Thirty and six rather than 30 and 6

LINE 85-95 needs to be in methods

Line 102 number of results?  more details on number of people who performed interviews.  Did anyone refuse?  How were the individuals selected to interview? Translations?

107 widespread

109 remove However

Table 4 please describe in text the definition for patient agency

Line 118  how did you decide to pick the 3 patient, medicine, context

Discussion Line 137 this is hypothesis generating and I would avoid using terms such as "it seems evident"

Line 143-`145  Does this broaden the scope--what evidence to determine the 3 aspects

Line 159  this is the first mention of utensils, should it be used before? 

Section 4.2 has numerous errors in grammar and has too many thats.  Avoid using terms such as Having said that, Nevertheless, but also, in consideration of generally   

Line 221  remove In fact

Line 236  Nineteen rather than 19

Line 244  start with Reasons for non-adherence were...

Could you add a #5 under 5.4  providing dosing spoons

Line 280  change to It may contribute to some of ....

Research limitations need to be expanded, others need to be added for example low numbers of interviews especially within certain countries.  not all countries are sub-Saharan.  Training of interviewers, consistencies?

Line 299-300 delete ...to that which has been gained in this research.

Line 303 stop-smoking course requiring adherence. 

Line 312 either reword the Ideas for this can be found in [21} or delete. It is out of place here (why not in discussion or introduction?)

Comments on the Quality of English Language

Needs some significant improvement, Some medications in BAN versus USAN vs INN, depends on what style you want.  

Author Response

Thank you for your detailed analysis of my paper. I have made the changes you request. I have annotated your analysis in the attached document. I trust that my responses to your analysis are satisfactory.

Round 2

Reviewer 1 Report

Comments and Suggestions for Authors

It is recommended to gather data of some more persons from countries where you have only one data point for better significance

Comments on the Quality of English Language

Minor review of English language needed to refine the manuscript

Author Response

I appreciate your suggestion to gather more data, but this is no longer possible. The purpose of the research was not to reach "significance" since it is not quantitative research, but to gain qualitative insights from as wide a range of people as feasible at that time. Gathering data across so many countries was valuable in itself, I believe.

I have reviewed the English once again.

Reviewer 3 Report

Comments and Suggestions for Authors

Thank you for thoughtful consideration of the suggested changes. 

Comments on the Quality of English Language

Minor 

Author Response

Thank you again for your review. I appreciate your contribution to improving the quality of the paper.